# The Physiological Roles of Vitamin E and Hypovitaminosis E in the Transition Period of High-Yielding Dairy Cows

**DOI:** 10.3390/ani11041088

**Published:** 2021-04-11

**Authors:** Satoshi Haga, Hiroshi Ishizaki, Sanggun Roh

**Affiliations:** 1Division of Grassland Farming, Institute of Livestock and Grassland Science, NARO, Nasushiobara, Tochigi 329-2793, Japan; hishizak@affrc.go.jp; 2Graduate School of Agricultural Science, Tohoku University, Sendai 980-8572, Japan; sanggun.roh.a3@tohoku.ac.jp

**Keywords:** alpha-tocopherol/vitamin E-related gene, calving, colostrum, high-yield dairy cows, inflammation, health, lactation, liver, mammary gland, oxidative stress

## Abstract

**Simple Summary:**

In high-yield cows, most production diseases occur during transition periods. Alpha-tocopherol, the most biologically active form of vitamin E, declines in blood and reaches the lowest levels (hypovitaminosis E) around calving. Hypovitaminosis E is associated with the incidence of peripartum diseases. Therefore, many studies which have been published for more than 30 years have investigated the effects of α-tocopherol supplementation. This α-tocopherol deficiency was thought to be caused by complex factors. However, until recently, the physiological factors or pathways underlying hypovitaminosis E in the transition period have been poorly understood. In the last 10 years, the α-tocopherol-related genes expression, which regulate the metabolism, transportation, and tissue distribution of α-tocopherol in humans and rodents, has been reported in ruminant tissues. In this paper, we discuss at least six physiological phenomena that occur during the transition period and may be candidate factors predisposing to a decreased blood α-tocopherol level and hypovitaminosis E with changes in α-tocopherol-related genes expression.

**Abstract:**

Levels of alpha-tocopherol (α-Toc) decline gradually in blood throughout prepartum, reaching lowest levels (hypovitaminosis E) around calving. Despite numerous reports about the disease risk in hypovitaminosis E and the effect of α-Toc supplementation on the health of transition dairy cows, its risk and supplemental effects are controversial. Here, we present some novel data about the disease risk of hypovitaminosis E and the effects of α-Toc supplementation in transition dairy cows. These data strongly demonstrate that hypovitaminosis E is a risk factor for the occurrence of peripartum disease. Furthermore, a study on the effectiveness of using serum vitamin levels as biomarkers to predict disease in dairy cows was reported, and a rapid field test for measuring vitamin levels was developed. By contrast, evidence for how hypovitaminosis E occurred during the transition period was scarce until the 2010s. Pioneering studies conducted with humans and rodents have identified and characterised some α-Toc-related proteins, molecular players involved in α-Toc regulation followed by a study in ruminants from the 2010s. Based on recent literature, the six physiological factors: (1) the decline in α-Toc intake from the close-up period; (2) changes in the digestive and absorptive functions of α-Toc; (3) the decline in plasma high-density lipoprotein as an α-Toc carrier; (4) increasing oxidative stress and consumption of α-Toc; (5) decreasing hepatic α-Toc transfer to circulation; and (6) increasing mammary α-Toc transfer from blood to colostrum, may be involved in α-Toc deficiency during the transition period. However, the mechanisms and pathways are poorly understood, and further studies are needed to understand the physiological role of α-Toc-related molecules in cattle. Understanding the molecular mechanisms underlying hypovitaminosis E will contribute to the prevention of peripartum disease and high performance in dairy cows.

## 1. Introduction

Calving is an unavoidable event in the production of milk and young cattle in dairy farms. However, during the transition period (from three weeks before calving to three weeks after calving), high-yield dairy cows experience severe energy and nutrient deficiencies [1,2], metabolic and endocrine changes [3,4], peripartum stress [2], and inflammation [5]. This leads to an imbalance between pro-oxidants and antioxidants, eventually resulting in oxidative stress [6] and immune dysfunction [7], which increases the risk of peripartum diseases. Indeed, most production diseases (mastitis, ketosis, digestive disorders, and laminitis) occur before and soon after calving [8,9].

Vitamin E (VE) has essential antioxidant functions and is an important nutrient for cows. It is well known that blood VE levels decline gradually throughout prepartum, reaching the lowest levels (hypovitaminosis E) after delivery [2,10,11,12,13]. Several studies have documented that lower blood VE concentrations are associated with the incidence of peripartum diseases such as mastitis [14], retained foetal membranes [15] and left displaced abomasum [16]. The effect of VE supplementation, as a practical measure to counter hypovitaminosis E and the high risk of peripartum diseases, have been often controversial in published literature for more than 30 years. To the best of our knowledge, many detailed reviews reported in the last 30 years, focused primarily on the disease risk and the effect of VE supplementation on the health of dairy cows and heifers in transition period (two databases, PubMed and Web of Science were searched using key words: vitamin E, dairy cows, tocopherol, 1990–2021, and reviews were picked up whose full-text is currently available online; 1990s: [17,18,19,20,21,22], 2000s: [1,23,24,25,26,27], 2010s: [28,29,30,31,32]). This VE deficiency may be caused by complex factors, such as changes in VE intake and its transfer into colostrum around calving [26]. However, the physiological factors underlying hypovitaminosis E in the transition period of high-yielding dairy cows were less well understood until recently. In the last 10 years, several studies have shown evidence that the ovine [33,34] and bovine [35] liver may play an important role in the regulation of VE disposition because of the high expression of VE-related molecules. In addition, other peripheral tissues showed unique expression patterns for VE-related molecules and VE accumulation properties in cattle [2,35,36,37,38]. These findings indicate that the expression of VE-related genes in the liver and non-hepatic tissues may be involved in the regulation of VE status in cows. Therefore, it is necessary to explore and discuss the physiological factors underlying hypovitaminosis E in dairy cows based on the latest reports on the expression of VE-related genes in bovine tissues. Especially during peripartum period in dairy cows, dramatic changes in lipid metabolism [39], physiological stress and inflammation may lead to hepatic injury, dysfunctions [40] and hepatocyte apoptosis [41]. Furthermore, during the onset of lactation, the mammary gland undergoes dramatic functional and metabolic changes during the transition period [42]. Novel knowledge about the changes in the expression of VE-related genes in the liver and mammary glands of transition dairy cows is compelling.

Thus, the current study aimed to understand the occurrence of hypovitaminosis E during the transition period, and contribute to the development of an effective feeding system for the health and high performance of dairy cows. Therefore, the aims of this review are: (1) to summarise the basic information about VE and VE-related molecules from the latest literature; (2) to re-evaluate the physiological roles of VE and the relationship between the risk of peripartum diseases and hypovitaminosis E in transition dairy cows; and (3) to discuss the physiological factors underlying hypovitaminosis E in relation to changes in the VE-related genes expression levels, especially in the liver and mammary gland from late pregnancy to early lactation.

## 2. Vitamin E and VE-Related Molecules

### 2.1. Vitamin E (α-Tocopherol)

In 1922, Evans and Bishop [43] discovered VE as an essential micronutrient for reproduction in rats. Subsequently, considerable research has been conducted on VE function and metabolism globally during the last one-hundred years. Vitamin E is considered the most effective, fat-soluble trace compound and chain-breaking antioxidant, which protects cell membranes from peroxidative damage [44]. It also plays specific roles beyond its antioxidant function, such as cellular signalling and regulation of gene expression [45,46]. The clinical importance of VE is significantly increasing to prevent various diseases in humans; VE deficiency can lead to neurological abnormalities, such as ataxia [47] and blindness [48]. The importance is also well recognised in animal production and medicine to maintain the health of livestock. Vitamin E cannot be synthesised in the mammalian body and must therefore be provided through food or supplementation. The VE family is composed of four tocopherols and four tocotrienols (α, β, γ and δ). Tocopherols have a saturated phytyl side chain, whereas tocotrienols have a three-fold unsaturated isoprenoid side chain [44]. Unlike other nutrients, the body cannot interconvert among these forms. In the plasma and tissues of humans and animals, one form, α-tocopherol (α-Toc), is the predominant congener and the most biologically active form of VE [44] because other tocopherols and tocotrienols are usually found at very low-levels compared to α-Toc. Alpha-tocopherol regulates key cellular events by mechanisms unrelated to its antioxidant properties, such as the inhibition of protein kinase C activity [44]. Furthermore, the expression of many genes has been found to be under the non-antioxidant control of α-Toc [45,49], suggesting the high availability of α-Toc in protection against disease in humans and animals.

Natural α-Toc (RRR-α-Toc (2,5,7,8-tetramethyl-2R-(4′R,8′R,12-trimethyltridecyl)-6-chromanol)) has the highest biological activity and is maintained at the highest level in plasma and tissues of humans and animals [50]. However, most of the α-Toc used for supplementation of food and feed is synthetic in origin, designated as *all*-*rac*-α-Toc (2,5,7,8-tetramethyl-2RS-(4′RS,8′RS,12-trimethyltridecyl)-6-chromanol). *All*-*rac*-α-Toc (*dl*-α-Toc) has three asymmetric carbons at positions 2, 4′ and 8′ and consists of an equimolar mixture of eight stereoisomers (RRR, RRS, RSS, RSR, SRR, SSR, SRS and SSS). The measurement of α-Toc activity in terms of IU was based on fertility enhancement by the prevention of spontaneous abortions in pregnant rats [51]; 1 IU of α-Toc is defined as 1 mg of *all*-*rac*-α-tocopheryl acetate, as 0.74 mg of RRR-α-tocopheryl acetate, and as 0.67 mg of RRR-α-Toc. Data from cows comparing the bioavailability of various Toc stereoisomers are contradictory, and insufficient consistent data are available to determine IU conversion factors for VE for ruminants [52]. However, Meglia et al. [53] showed that RRR-α-Toc was the most predominant stereoisomer, constituting more than 86%, whereas the remaining part of α-Toc was made up of three synthetic 2R isomers, while the 2S isomers contributed less than 1% of the total α-Toc in plasma and milk from dairy cows supplemented with *all*-*rac*-α-tocopheryl acetate. Jensen et al. [54] showed that after a single dose injection of *all*-*rac*-α-tocopheryl acetate, the RRR-α-Toc was retained in plasma for the longest time and secreted into milk at the highest concentration followed by RRS-, RSS-, and RSR-α-Toc leaving the Ʃ2S-α-Toc to be retained in plasma for the shortest time and secreted into milk at the lowest concentration. In dairy cows during early lactation, the serum concentrations (nM) of γ-Toc, α-, β-, γ- and δ-tocotrienol were far lower than those of α-Toc (approximately 1/56, 1/214, 1/3947, 1/5000 and 1/2500, respectively) [55]. Among the naturally occurring forms of the VE family, α-Toc only meets VE requirements and α-Toc stereoisomers have different bioactivities because α-Toc transfer protein (αTTP) has a different affinity for VE (described in Section 2.2.1) and play an important role in the circulation and disposition of α-Toc in cattle. Throughout this review, the terms “VE” and “α-Toc” are used interchangeably and the plasma/serum concentrations of α-Toc is unified and expressed in “μg/mL” (1 μM = 0.43 μg/mL).

### 2.2. Alpha-Tocopherol Transfer Protein and Other α-Toc-Related Molecules

Alpha-tocopherol shows tissue-specific distribution in animals [56], and this property may affect the α-Toc potencies for each tissue. However, until recently, the molecular mechanisms underlying this tissue distribution and the action of α-Toc were poorly understood in cattle. Pioneering studies from the 1990s conducted with humans and rodents have identified and characterised some α-Toc-related proteins, molecular players involved in α-Toc regulation, followed by a study in ruminants from the 2010s.

#### 2.2.1. Alpha-Tocopherol Transfer Protein

A cytosolic protein that specifically binds to α-Toc was purified from rat and human liver [57], and the full-length cDNA sequence of the rat and human homolog has been reported [58,59]. The protein is called α-Toc transfer protein (αTTP) encoded by the *TTPA* gene, which is classified as a member of the Sec14 like protein family, which has a CRAL-TRIO lipid-binding domain. Humans carrying mutations in the *TTPA* gene revealed a low α-Toc level in plasma and neurological disorders associated with elevated oxidative stress termed as ataxia with VE deficiency (AVED), indicating the importance of αTTP in regulating plasma α-Toc levels [60,61]. In agreement with this fact, αTTP^-/-^ mice demonstrated AVED-like symptoms [48]. Arita et al. [62] reported that α-Toc secretion was markedly stimulated when αTTP was overexpressed in a cultured rat hepatocyte cell line. These reports indicate that hepatic αTTP regulates α-Toc secretion from the liver into the circulation. In intracellular transfer mechanisms, αTTP binds to α-Toc in the endosomal membrane and enables its transport to the plasma membrane, where the αTTP interacts with phosphatidylinositol-4,5-bisphosphate [PI(4,5)P2] or [PI(3,4)P2] for the release of α-Toc and its incorporation into the plasma membrane [63]. Mutations in the *TTPA* prevent its binding with the membrane PIPs and transfer α-Toc to the membrane [63,64,65]. Alpha-tocopherol transfer protein translocate to the hepatic endosomal compartment to transfer available α-Toc [66,67,68], and the intracellular localisation of αTTP in hepatocytes is dynamic and responds to the presence of α-Toc [67]. Moreover, the regulation of *TTPA* expression in tissues in response to α-Toc supplementation is unclear, with complicated reports on rodents genes. Concerning the effect on *TTPA* mRNA expression in the liver, there were conflicting reports that α-Toc supplementation in rats was up-regulated [69], down-regulated [70] and showed no [71] effective actions on gene expression. Recently, higher expression of *TTPA* mRNA in chicken liver, in response to dietary α-Toc content, was reported, and the result may suggest its crucial role in the transport of α-Toc in chicken liver [72]. The αTTP, which is mainly expressed in the liver, regulates α-Toc secretion from the liver into circulation, and is also observed in other peripheral tissues and/or cells, such as mouse uterus [73], human leukocytes [74], human placental trophoblast cells [75], mouse lung [76], chicken small intestine, intestinal mucosal layer and adipose tissues [72].

Molecular cloning and characterisation of the full-length cDNA of ovine and bovine *TTPA* genes were conducted [33,77], which contained 2740 nucleotides, and the open reading frame contained 846 bp encoding 282 amino acids with 88% identity with the human genes. Bovine and ovine αTTP have an additional five amino acids (GEEVT) at the C terminus, which goat, bison, deer, dolphin, and killer whale αTTPs contain, whereas human, mouse, and rat αTTPs do not have, suggesting that the C-terminal sequences of αTTP are specific to *Cetartiodactyla* animals [77]. Zuo et al. [78] demonstrated the expression of *TTPA* mRNA in non-hepatic ovine tissues, including the heart, spleen, lung, kidney, and muscle. Haga et al. [35] reported the distribution of *TTPA* mRNA expression in 20 major tissues in calves, including metabolic, reproductive, endocrine, immune, digestive, and absorptive tissues. Furthermore, the hepatic [2,55,79] and mammary [2] *TTPA* mRNA expression in dairy cows was also investigated (described in Section 4). These reports suggest that αTTP is expressed not only in the liver, but also in various non-hepatic tissues in cattle, and may also play a crucial role in regulating α-Toc circulation and local α-Toc status.

#### 2.2.2. Afamin

Another potential candidate protein for α-Toc binding and transport to plasma and extravascular fluids is afamin (AFM) encoded by the *AFM* gene [80,81]. Afamin belongs to the albumin (*ALB*) gene superfamily, which comprises ALB, α-fetoprotein, and vitamin D-binding protein [82] and has multiple binding sites for both α-Toc and γ-Toc [80]. Afamin is primarily expressed in the liver [82] and secreted into circulation, where it is partially associated with apolipoprotein A1 (ApoA1), which contains high density lipoprotein (HDL) subfractions in human plasma [81]. In women, nearly a two-fold increase in serum AFM concentrations was observed during uncomplicated pregnancy [83] and the elevated serum concentrations were associated with the presence of metabolic syndrome [84]. These findings suggest that plasma/serum AFM concentrations have the potential to serve as predictive markers for various medical conditions. However, AFM and α-Toc concentrations are significantly correlated in follicular and cerebrospinal fluids, but not in plasma or serum [81,85,86]. The expression of AFM has also been confirmed in other organs such as the human kidney [83], and Kratzer et al. [87] reported that AFM is synthesised by brain capillary endothelial cells and mediates α-Toc transport into the central nervous system across the blood-brain barrier. According to these reports, AFM might be a specific binding/transport protein contributing to α-Toc circulation and its status in various local tissues.

Haga et al. [35] reported the distribution of *AFM* mRNA expression in the liver, renal cortex, testis, thymus, duodenum, and jejunum tissues of calves. Although hepatic and mammary *AFM* mRNA expression in dairy cows was investigated, the *AFM* transcript in mammary gland tissue was not detected [2]. The understanding of the physiological role of AFM in ruminants is minimal. However, in dairy cows, 3% of plasma α-Toc was not associated with the lipoprotein fractions in circulation [88]; thus, AFM might be involved in α-Toc transportation in plasma and extravascular fluids.

#### 2.2.3. Tocopherol-Associated Protein

Studies have reported the identification of tocopherol-associated protein (TAP/*SEC14L2*) in the cytosol of bovine liver, and TAP has a sequence that is homologous to the proteins with the CRAL-TRIO structural motif in common with αTTP [89,90]. Recombinant human TAP could bind only to α-Toc but not to other tocopherols, as shown by ligand competition analysis and α-Toc-dependent nuclear translocation and transcriptional activation properties in transfected COS-7 cells [91]. This report suggested that TAP might be associated with intracellular metabolism, non-antioxidative function, and the regulation of gene expression of α-Toc [92]. However, TAP, expressed in mouse mast cells, was predominantly localised in the cytoplasm and its subcellular localisation was not changed by α-Toc [93]. These results suggest that the physiological role of TAP in mast cells is not α-Toc-related, while as an α-Toc binding protein, TAP can promote α-Toc retention and thus increase its concentration in breast cancer cells [94]. *SEC14L2* mRNA has also been observed in various human tissues [89,90]; however, the biological roles of TAP in each tissue are still poorly understood.

Haga et al. [35] reported the distribution of bovine *SEC14L2* mRNA expression in 20 major tissues of calves. Furthermore, the hepatic [2,55] and mammary [2] *SEC14L2* mRNA expression in dairy cows was also investigated (described in Section 4).

#### 2.2.4. Scavenger Receptor Class B, Type I

Some lipoprotein receptors and transporters might also be important for the control of α-Toc distribution in tissues because HDL, low density lipoprotein (LDL), very low-density lipoprotein (VLDL), and chylomicron (CM) are the major carriers of α-Toc in the bloodstream because of the hydrophobic properties of α-Toc [95]. Several studies using knock-out mice and over-expressing cells [95], and specific antibodies and a chemical inhibitor in enterocytes [96], have suggested that selective cholesterol ester uptake from HDL by scavenger receptor class B, Type I (SRBI), rather than endocytosis (Ex. VLDLs and LDLs), are important factors for α-Toc delivery into cells. Scavenger receptor class B, Type I, encoded by the *SCARB1* gene, is a member of a multiligand family that plays a well-established role as an HDL receptor [97]. In SRBI-deficient mutant mice, there was a significant increase in plasma α-Toc that was mostly distributed in HDL-like particles and a significant decrease in the α-Toc concentrations in bile and several tissues, including ovary, testis, lung, and brain, but not in the liver, spleen, kidney, or white fat [95]. These reports suggest that SRBI plays an important role in transferring α-Toc from plasma lipoproteins to specific tissues and the nervous system [98]. In addition, SRBI was also shown to mediate α-Toc efflux from the cytosolic compartment of Caco-2 cells to the apical medium, suggesting a potential regulatory role in α-Toc absorption [96].

In cattle, HDL is the major lipoprotein in the plasma and follicular fluid [99,100], and α-Toc is mainly located in HDL among lipoproteins [37]. Rajapaksha et al. [101] sequenced bovine *SCARB1* cDNA, which contains 509 amino acids. The changes in *SCARB1* mRNA levels were evaluated in developing bovine ovarian cells; however, the relationship between the mRNA level and α-Toc concentration in follicular fluid is unknown [99,100]. By contrast, Higuchi et al. [37] clarified that the upregulation of *SCARB1* mRNA in neutrophils in cattle supplemented with α-Toc and the cellular α-Toc contents were decreased after anti-SRBI treatment. These results suggest that SRBI is a crucial receptor in bovine neutrophils for the uptake of HDL-associated α-Toc. A study investigating the distribution of *SCARB1* mRNA in six tissues from cows demonstrated that their levels were high in the adrenal cortex and corpus luteum [101] because these organs take up large amounts of cholesterol from the bloodstream HDL to synthesise steroid hormones. Haga et al. [35] also reported high α-Toc accumulation in the adrenal gland and testis, with the highest expression levels of *SCARB1* mRNA among the 20 tissues in calves. These results suggest that the high expression of SRBI in these tissues may take up some α-Toc along with HDL.

#### 2.2.5. ATP-Binding Cassette Transporter A1

ATP-binding cassette transporter A1 (ABCA1/*ABCA1*) is a cholesterol efflux regulatory protein. It is known that ABCA1 is involved in the regulation of cholesterol efflux from cells, and mutations in ABCA1 genes cause HDL deficiency [102]. In hepatocytes, lipid-free apoA1 is secreted to ABCA1, which localises on the plasma membrane and into intracellular sites, and nascent HDL (preβ-HDL) particles are formed [103,104]. It was also reported that ABCA1 mediates cellular secretion of α-Toc because hepatic α-Toc secretion is suppressed by ABCA1-RNAi or probucol (inactivator of ABCA1) in a rat hepatoma cell line and C57BL/6Cr mice in vivo [105]. Kono and Arai [66] demonstrated that αTTP transports α-Toc to the plasma membrane, where it is picked up by ABCA1 and excreted from the hepatocyte. The expression profile of human *ABCA1* mRNA in different tissues has been previously reported [106,107]. In non-hepatic cells, forced expression of ABCA1 markedly stimulated α-Toc efflux in baby hamster kidney cells [108]. Therefore, the regulation of the transportation and distribution of α-Toc must be closely linked to the complex mechanisms of cholesterol, lipoprotein and especially HDL metabolism via ABCA1.

Sequence analysis of bovine *ABCA1* cDNA revealed that the open reading frame of this gene consists of 6786 bases and encodes a protein of 2261 AA with a predicted molecular weight of 254 kDa [109]. Haga et al. [35] reported that the *ABCA1* mRNA level in Japanese Black beef calves was the highest in the liver, followed by heart muscle, lung, adipose, spleen and adrenal gland, which did not agree with the report of Farke et al. [109], who detected the highest mRNA level in the lungs of an adult lactating Holstein–Friesian cow. This discrepancy might be attributable to the differences in the bovine breeds and the life stages of the animals because of the different lipid metabolisms. In particular, the differences in cholesterol status may affect the distribution of *ABCA1* mRNA expression in tissues because *ABCA1* transcript activity is reportedly regulated by the liver X receptor and sterol regulatory element-binding protein (SREBP) 2, which are key proteins in cholesterol metabolism [102,110]. Hepatic gene expression in transition dairy cows has been reported in recent studies [2,111,112]. Furthermore, it is notable that the expression and localisation of ABCA1 in the bovine mammary epithelial cells, mammary gland, and milk fat globules have an important role in cholesterol homeostasis and milk fat synthesis [36,112,113,114,115,116]. The mammary expression of ABCA1 may be involved in the regulation and mechanism of α-Toc transfer into colostrum and milk.

#### 2.2.6. Cytochrome P450 Family 4, Subfamily F, Polypeptide 2

Cytochrome P450 family 4, subfamily F, polypeptide 2 (CYP4F2/*CYP4F2*), is a member of the CYP4F subfamily ω-hydroxylate leukotriene B_4_ [117,118,119]. In addition, tocopherols and tocotrienols are also metabolised by side chain degradation initiated by CYP4F2-catalyzed ω-hydroxylation, followed by β-oxidation, mainly in the liver [120]. The resulting water-soluble metabolites, carboxyethyl hydroxychromans (CEHC), are excreted in the urine [121,122]. Cytochrome P450 family 4, subfamily F, polypeptide 2 exhibited markedly higher catalytic activities for γ-Toc than α-Toc, resulting in preferential physiological retention of α-Toc and elimination of γ-Toc [120]. In particular, sesamin potently inhibited tocopherol-ω-hydroxylase activity exhibited by CYP4F2 [120], and dietary sesame seeds elevated α-Toc concentrations in the brain, liver and serum, and lowered the oxidative stress marker, thiobarbituric acid reactive substance (TBARS), in the brain of rats [123]. These data also emphasise the importance of CYP4F2 in α-Toc metabolism. The α-Toc levels in the body may be influenced by changes in the mRNA expression and enzyme activity of CYP4F2. In fact, the reduced expression of hepatic *TTPA*, *AFM* and *CYP4F2* genes probably leads to decreased plasma α-Toc levels and elevated α-Toc levels in the liver of streptozotocin-induced type 1 diabetes rat models [124]. Sterol regulatory element-binding proteins can transactivate *CYP4F2* transcription in hepatocytes [125], and decrease SREBP-1 proteins expression, resulting in reduced expression of CYP4F2, which slows the breakdown of α-Toc in experimental non-alcoholic fatty liver disease model mice [126].

In calves, the *CYP4F2* mRNA level was the highest in the liver, followed by the testis, adrenal gland, duodenum, and jejunum, which have high α-Toc accumulation [35]. The *CYP4F2* mRNA in lactating Holstein cows was significantly higher in the kidney than in the liver, lung, mammary gland, heart, skeletal muscle, spleen and uterus [38]. Furthermore, the hepatic [2,55] and mammary [2] *CYP4F2* mRNA expression in transition dairy cows was investigated (described in Section 4). The contribution of CYP4F2 to circulating α-Toc concentrations in transition dairy cows is not yet well understood. However, it is believed that the evidence of α-Toc metabolism by CYP4F2, similar to the metabolism of polyunsaturated fatty acids, can provide information regarding the physiological factors underlying hypovitaminosis E and the importance of CYP4F2 in the maintenance of dairy cow health [127].

As described above, in bovine species, the expression of *TTPA*, *AFM*, *SCARB1*, *ABCA1*, *SEC14L2* and *CYP4F2* genes in various tissues may play important roles in the regulation of α-Toc disposition (metabolism, transportation, and tissue distribution) (Figure 1). These genes may not be the whole explanation of α-Toc disposition mechanism [128,129,130]; however, the evidence of the expression of these α-Toc-related genes has the potential to help understand the physiological factors underlying hypovitaminosis E in the transition period of high-yielding dairy cows.

## 3. Hypovitaminosis E in Transition High-Yield Dairy Cows

### 3.1. Changes in α-Tocopherol Status in Transition Dairy Cows

It is well known that the plasma/serum concentrations of α-Toc in high-yield dairy cows gradually decrease throughout prepartum, starting from several weeks before calving, reaching a nadir at calving, and remaining at lower levels during the puerperal period (about 3–7 d), and increasing thereafter [2,10,11,12,13]. According to the fundamental research conducted by Weiss in 1990s [11,12,21,131,132,133], others [10,134] and NRC [52], based on disease risk and immune function in dairy cows, plasma/serum concentrations of α-Toc should be more than approximately 3 μg/mL in peripartum period; below this cut-off level is an α-Toc deficiency namely hypovitaminosis E. From 2000s to 2020s, the occurrence of hypovitaminosis E around calving has been still reported [2,13,15,16,79,135,136,137,138]. According to the NRC [52] recommendation to maintain this cut-off value, dry cows and heifers fed stored forages during the last 60 d of gestation require approximately 1.6 IU of supplemental α-Toc/kg BW (1120 IU/d for cows weighing 700 kg BW or approximately 80 IU/kg of DMI). However, the effect of supplementation and optimal dose may be far from certain. In cows supplemented with 1000 IU α-tocopheryl acetate per day (approximately 108 IU/kg of DMI) from 30 d prepartum to two weeks postpartum, the occurrence of hypovitaminosis E around calving was observed but with a lower decrease in plasma α-Toc concentration. However, when compared to the group with no α-Toc supplementation, the plasma α-Toc concentration was regained earlier after calving [135]. Hypovitaminosis E after calving also occurred in cows that were administered 1000 IU α-tocopheryl acetate per day (approximately 110 IU/kg of DMI) from 60 d prepartum to calving [139]. In agreement with these reports, the serum α-Toc concentration in high-yield dairy cows fed approximately 97 mg of α-toc/kg of DM from five weeks prepartum to calving, was less than 3 μg/mL during the prepartum period and approximately 1.5 μg/mL after calving [2]. By contrast, high-dose VE supplementation (3000 IU/day/cow) from eight weeks before the predicted calving date was sufficiently high to prevent hypovitaminosis E [140].

### 3.2. Disease Risk in Hypovitaminosis E and the Effects of α-Toc Supplemantetion in Transition Dairy Cows

To our knowledge, numerous detailed reviews about the disease risk of hypovitaminosis E and the α-Toc supplementation effect on the health of transition dairy cows and heifers have been published in the last 30 years (two databases, PubMed and Web of Science were searched using key words: vitamin E, dairy cows, tocopherol, 1990–2021, and reviewed full text is currently available online, 1990s: [17,18,19,20,21,22], 2000s: [1,23,24,25,26,27], 2010s: [28,29,30,31,32]). Based on three recent reviews published in the last 10 years [30,31,32], α-Toc supplementation has the potential to affect the incidence of mastitis, including milk somatic cell count (SCC) values, and retained foetal membranes (RFM). Some studies suggest that α-Toc supplementation at the level 1000 to 4000 IU/day/cow during the dry period can reduce the frequency of intramammary infection and the occurrence of clinical mastitis, as well as the levels of SCC in milk [17,133,141], suggesting that α-Toc deficiency may be a critical risk factor for the increased frequency of infection and duration in mammary glands during the transition period. Many studies have provided evidence suggesting that α-Toc supplementation can mitigate the immune dysfunction that occurs during the transition period [21,142,143,144,145,146]. The overproduction of reactive oxygen species (ROS) may contribute to several metabolic disturbances, resulting in the appearance of RFM [147]. The meta-analysis [148], performed to consolidate the results of studies that have evaluated the effect of α-Toc supplementation during the dry period on the incidence of RFM and found that α-Toc supplementation was associated with a decrease in the incidence of RFM. However, there are insufficient studies which have evaluated the effect of α-Toc alone on RFM, and much of the apparent significant benefit of α-Toc may be mediated by Se. The results of published studies were equivocal, and further work is required to assess α-Toc supplementation.

Next, we focus on relevant papers published in the last 10 years and present some novel data about the disease risk of hypovitaminosis E and α-Toc supplementation effects in transition dairy cows.

#### 3.2.1. Left Displaced Abomasum

Left displaced abomasum (LDA) occurs in multiparous cows during the first month of lactation as part of the peripartal disease complex. Hasanpour et al. [149] reported that cows with LDA had approximately 42% lower serum α-Toc concentrations than healthy cows as control (2.7 vs. 4.7 µg/mL, respectively), and suggested use of supplementary α-Toc with Se for the LDA cattle. Qu et al. [16] investigated a retrospective case-control study to determine whether the lower serum α-Toc concentrations precede or remain after LDA in the transition period. Seven multiparous Holstein cows diagnosed with LDA between days 6 and 32 postpartum and 10 healthy cows from the same herd were analysed. Before calving, all 17 cows were visually healthy. These cows fed a TMR contained supplemental *all*-*rac*-α-tocopheryl acetate at 167 and 24.5 IU/kg DM before and after calving, respectively. Each cow in LDA had other diseases before LDA diagnosis (five cows had ketosis, four cows had metritis, two cows had milk fever, and one cow each retained placenta after twins, mastitis, or laminitis). Most cows had diseases after LDA correction (until day 49 postpartum: four cows had ketosis, one cow had mastitis, and one cow died from an intestinal ulcer 2 d after LDA diagnosis at day 34 postpartum). Serum α-Toc concentrations decreased dramatically in the first week postpartum in all cows; however, the nadir α-Toc concentration at day 7 postpartum in LDA cows was lower than that in controls, and the level (about 2.2 µg/mL) was diagnosed as hypovitaminosis E. Furthermore, cows with LDA during the first month postpartum (for 28 days) had, on average, lower serum α-Toc concentrations than the cut-off value. In control cows, α-Toc concentrations returned to prepartal concentrations (approximately 6.0 µg/mL) by four weeks postpartum, while serum α-Toc in cows with LDA remained lower than controls during the entire postpartum sampling period (seven weeks postpartum). A lower DMI might be a probable causative factor for the lower serum α-Toc concentrations in LDA cows, however, there was an absence of DMI data in the study [16]. The authors suggested that these findings indicate that lower serum α-Toc concentrations are a potential early indicator for the development of LDA in multiparous cows.

#### 3.2.2. Retained Foetal Membranes, Stillbirth, and Reproductive Performance

Retained foetal membranes are an established risk factor for other peripartum diseases and fertility. Multiple physical, endocrine, and cellular factors are involved in RFM, and the immune and antioxidant potential before calving are important predisposing factors. Qu et al. [15] investigated the risk indicators of RFM using a nested case-control design and compared multiparous dairy cows that developed RFM with cows that remained healthy or cows that developed other diseases (metritis, mastitis, ketosis, or laminitis) in early lactation (each n = 32). These cows fed a TMR contained supplemental VE at 167 and 24.5 IU/kg DM as *all*-*rac*-α-tocopheryl acetate before and after calving, respectively. During the three weeks pre-calving, RFM cows had lower serum α-Toc concentrations than healthy cows. In addition, RFM cows tended to have lower serum α-Toc concentrations between three and two weeks prepartum than cows that developed other diseases. After calving, the α-Toc concentrations in RFM and other diseased cows were lower than the cut-off values (<3 µg/mL). Pontes et al. [150] evaluated the effects of injectable α-Toc supplementation during the last three weeks prepartum on the incidence of RFM and stillbirth, and reproductive performance in cows fed limited amounts of dietary α-Toc. During the prepartum period, cows were fed with less than 500 IU of supplemental *dl*-α-tocopherol per day. Cows were randomly assigned to remain as untreated controls (n = 441) or to receive three intramuscular injections of 1000 IU each of *dl*-α-tocopherol administered at three, two, and one week before calving (VitE group, n = 449). The serum α-Toc concentration at three weeks prepartum was similar between the control (n = 75) and VitE (n = 66), with an average of 2.97 μg/mL. The results showed that treatment with injectable α-Toc decreased the RFM rate from 20.1% to 13.5% and, decreased the incidence of stillbirth from 14.9% to 6.8%. In addition, cortisol is known to suppress leukocyte function in cattle, and its serum levels in cows receiving α-Toc were lower than those in control cows at 1 week before calving. The authors suggested that improved immune-cell function, through either antioxidant effects or other cellular signalling pathways activated by α-Toc, is likely to underlie the reduction in the incidence of RFM. Cows with VitE also tended to have improved pregnancy per insemination at first AI (36.7 vs. 30.1%) because of decreased pregnancy loss than control cows. Despite a similar insemination rate, VitE cows had a 22% higher pregnancy rate than control cows. The authors concluded that feeding dairy cows with less than 500 IU of VE per day during the last weeks of gestation may result in an inadequate level of plasma α-Toc that leads to compromised-peripartum health and subsequent reproduction.

#### 3.2.3. Udder Health (Mastitis and SCC Values) and Milk Yield

Politis et al. [14] investigated the relationship between the incidence of clinical mastitis and blood α-Toc levels during dry off and calving. All cows (n = 146) were supplemented with *all*-*rac*-α-Toc at a rate of 3000 and 50 IU/cow per day during the dry period and lactation, respectively. According to the blood α-Toc concentrations, three groups at dry off were created: high (>6.25 μg/mL), medium (4.25–6.25 μg/mL), and low (<4.25 μg/mL). In addition, three groups at calving were created: high (>3 μg/mL), medium (2–3 μg/mL), and low (<2 μg/mL). No differences were observed in the incidence of mastitis between the three α-Toc groups during the dry-off period, however, the incidence of mastitis was four times lower in the high- (>3 μg/mL) and medium- (2–3 μg/mL) α-Toc groups than that in the low-α-Toc group (<2 μg/mL) at calving. In fact, cows with mastitis had lower concentrations of α-Toc (1.9 μg/mL) than healthy cows (2.74 μg/mL) at calving. These results also suggest that supplementation with *all*-*rac*-α-Toc at a rate of 3000 IU/cow per day during the dry period cannot always prevent hypovitaminosis E around calving.

Several other factors affect α-Toc availability and its physiological functions in cows, including the source of the α-Toc active substance, other fat-soluble nutrients in feeds, timing and period of supplementation, inclusion of Se, α-Toc content of the basal feeds, and method of administration (e.g., ruminal pellets or premixes in diet, or iv, sc and im injection). Moghimi-Kandelousi et al. [151] evaluated the effects of α-Toc supplementation on the serum level, milk yield, and SCC values of transition cows by considering a large set of variables that might influence the responses to α-Toc supplementation. To conduct a comprehensive meta-analysis of α-Toc supplementation effects during the transition period, after a broad search in journals and databases with keywords related to transition cows supplemented with VE and appropriate filtering of the results, 36 papers including 53 trials were selected from 528 publications (from 1979 to 2018), and their data were extracted into a database. Overall, 22 studies were conducted on Holstein cows, and the rest used other breeds. In 10 studies, primiparous and multiparous cows were studied, whereas in the remainder, treatments were applied only to multiparous cows. Six papers (12 independent trials) were used in the meta-analysis of milk yield, in which eight trials reported positive effects of VE supplementation. Furthermore, meta-regression showed that breed, Se supplementation, number of days treated prepartum, parity, and method of administration did not alter the effect of VE administration on milk yield in the first month of lactation. By contrast, the overall results of the meta-analysis showed non-significant changes in milk SCC with VE supplementation. The meta-analysis and meta-regression also showed that VE supplementation improved the reproductive performance of transition cows, such as shorter days open, reduced the number of services per conception, and decreased the odds of RFM. In conclusion, the authors suggested that up to 3600 IU/day of VE as an oral supplement during the transition period affects the milk production and reproduction performance of cows with Se supplementation.

As mentioned above, relevant papers published in the last 10 years with novel data indicated that hypovitaminosis E in the transition period is a risk factor for peripartum disease and lower performance in dairy cows. Alpha-tocopherol supplementation is an important method for the effective prevention of peripartum disease in high-yield dairy cows, although supplementation during the dry period cannot always prevent hypovitaminosis E around calving.

## 4. Physiological Factors Underlying Decreased Blood α-Toc Level and Hypovitaminosis E in Transition Period

The α-Toc deficiency may be caused by complex factors such as changes in the amount of α-Toc intake, increased oxidative stress and lipid peroxidation, and transfer of α-Toc into colostrum around calving. However, until recently, the physiological factors or pathways underlying hypovitaminosis E in the transition period of high-yielding dairy cows have been less well understood. Dramatic changes in lipid metabolism [39], endocrine status [3,4], physiological stress [2] and inflammation [5] occur and may damage the hepatic functions [40,41], playing a crucial role in the metabolism and disposition of α-Toc [33,34,35] during peripartum in dairy cows. Therefore, we discuss the candidate physiological factors underlying decreased blood α-Toc levels and hypovitaminosis E during late pregnancy to early lactation period.

### 4.1. The Decline in α-Toc Intake by Decreasing DMI from Close-Up Period to Calving

In dairy cows, daily intake of DM gradually decreases during the dry period, especially during the close-up period, and the DMI drops sharply to a nadir level at calving and increases thereafter toward early lactation [53,152,153]. It is believed that the major cause of the decline in DMI may be reduced rumen volume and capacity beyond the space requirement of developing foetuses in the close-up period, and physical and physiological stress-induced parturition. Decline in DMI is an unavoidable physiological phenomenon in dairy cows. If the α-Toc content in the feed is not different during this period and no supplementation is provided, its intake amount decreases with the decline in DMI [2]. Thus, to determine whether the decrease in serum α-Toc concentrations reflects the decline in α-Toc intake due to decreasing DMI around calving, Haga et al. [2] compared the rate of change between the serum α-Toc concentrations and its intake using monitoring data from high-yield Holstein cows (n = 28). During the close-up period, the α-Toc intake gradually decreased, declining at calving. After calving, DMI and α-Toc intake progressively recovered and increased with time. The serum concentrations of α-Toc decreased during the close-up period, reaching a nadir after parturition until 0.5 week after parturition. A comparison of the changes in α-Toc intake levels and serum α-Toc concentrations around the calving period (−2 to 2 weeks relative to parturition) revealed that, in prepartum, the decreasing α-Toc rates were similar between intake and blood; however, after calving, increasing serum α-Toc levels were significantly delayed compared to the recovery of α-Toc intake. These results suggest that decreased α-Toc intake levels may be one of the causes that strongly influence the decrease in serum α-Toc concentrations until pre-calving, but not the only influencer during post-calving.

### 4.2. Changes in the Digestive and Absorptive Functions of α-Toc with Change in the Expression of α-Toc-Related Genes

It was reported that *all*-*rac*-α-tocopheryl acetate was stable in the rumen of high-yielding dairy cows [154]. However, these cows are at the risk of subacute ruminal acidosis (SARA; diagnosed when reticulo-ruminal pH is <5.6 for more than three hours per day) during the periparturient period [155]. The stability of α-Toc under SARA conditions is not well known.

Blood was sampled through a trial from sheep with a ligated pylorus that received α-Toc, suggesting that no significant amount of α-Toc was absorbed from the preintestine region to the blood stream in ruminants [156]. This finding was indirectly supported by Haga et al. [35], who investigated the expression of α-Toc-related genes and α-Toc accumulation in weaned calves with and without oral administration of α-Toc. These results indicated that, in the gastrointestinal (GI) tract (rumen to the colon), the jejunum and duodenum had high α-Toc content and expressed high levels of *SRB1*, *AFM*, *TAP* and *CYP4F2* mRNA. The functions of αTTP and TAP in the GI tract have not been clarified, however, these results suggest that the expression of α-Toc-related genes is involved in the regulation of absorbed dietary α-Toc in the GI tract. Thus, these small intestine regions may be the major oral α-Toc absorption sites in cattle. However, to the best of our knowledge, there has been no detailed investigation of the changes in the digestive and absorptive function of α-Toc in the GI tract of high-yield dairy cows during the transition period, although there may be a possibility of changing the function with changes in the expression of α-Toc-related genes. Although challenging, further studies investigating in detail about the changes in the digestive and absorptive function of α-Toc, with α-Toc-related genes expression, in the GI tract of high-yield dairy cows are needed.

### 4.3. The Decline of Plasma HDL Level as an α-Toc Carrier from Close-Up Period to Calving

Alpha-tocopherol, a fat-soluble vitamin, requires a carrier system for bloodstream transportation. Herdt and Smith [88] investigated the distribution of α-Toc and cholesterol among the various lipoprotein density fractions in the blood of lactating Holstein cows; the percentage of total plasma α-Toc and cholesterol were VLDL (2% and 2%), LDL (17% and 22%), and HDL (77% and 72%, respectively). In addition, the α-Toc:cholesterol ratios were not significantly different among the lipoprotein fractions. These results indicated that α-Toc and cholesterol were distributed in equal proportions among lipoprotein fractions and HDL is a major lipoprotein carrier of α-Toc in the plasma of dairy cows. These results were supported by Higuchi et al. [37]. During the transition period, plasma HDL, VLDL, LDL, and cholesterol levels in dairy cows gradually decreased throughout prepartum, reaching a nadir at calving, and increasing thereafter [2,112,157]. This decline in lipoproteins may be caused by (1) the reduced DMI intake [2,53,152,153]; (2) changes in lipid metabolism and increased plasma NEFA and BHBA [39,158]; (3) the impaired hepatic export mechanism with reduced secretion of ApoB100 (decreased mRNA) and apoA1 [159], and (4) enhanced transfer of these compounds into the fat rich-colostrum [112]. Based on the changes monitored in blood α-Toc/HDL ratio, which assesses the rate-limiting levels of HDL concentration as an α-Toc carrier, from −2 to 2 weeks relative to parturition, Haga et al. [2] confirmed that the α-Toc/HDL ratio was significantly lower during the post-calving period than during the pre-calving period. These results suggest that remaining at lower serum α-Toc concentrations after calving might not result in lower HDL concentrations. It appeared that other causes might determine the lower serum α-Toc concentrations during the first week after calving in addition to changes in the levels of α-Toc intake and plasma HDL concentrations.

### 4.4. Increasing Systemic Oxidative Stress and Consumption of α-Toc as Antioxidant around Calving

Oxidative stress in living organisms is generated when free radical production exceeds the capacity of antioxidant mechanisms. Considerable evidence [2,14,160,161,162,163] and reviews [6,164] suggest that high-yield dairy cows experience severe oxidative stress around calving and during the onset of lactation. Bernabucci et al. [165] reported that dairy cows with higher BCS and higher body condition losses are more prone to oxidative stress during the periparturient period. It was confirmed that the increase in systemic oxidative stress around calving roughly coincided with a decrease in blood α-Toc concentration [2]. Furthermore, mRNA expression of the major antioxidant enzymes in the liver was markedly downregulated at calving [2]. Since α-Toc is considered an important antioxidant, these results suggest that both systemic and hepatic antioxidative/oxidative balance may be lower, and the consumption of α-Toc increases around calving. In fact, supplementation with α-Toc could reduce the markers of oxidative damage, serum/plasma malondialdehyde (the product of lipid peroxidation) [139,166] and heat shock protein 70 [139].

### 4.5. Decreasing Hepatic α-Toc Transfer to Circulation with Change in the Expression of α-Toc-Related Genes

The study on bovine tissues distribution [35] demonstrated that the liver may play a central role in the regulation of α-Toc disposition, as inferred by the high hepatic expression of six α-Toc-related genes (see Section 2.2 about the information and references). However, high-yield dairy cows experience physiological stress [2], systemic inflammation [5], oxidative stress [139,160,167], hepatic endoplasmic reticulum (ER) stress [2,168], hepatocyte apoptosis [41], hepatic injury (necrosis-like cell death) [79] and development of fatty liver resulting in severe negative energy balance (NEB) because of high milk production immediately after calving [41,159,169]. Thus, liver function will be changed and substantially inhibited around calving.

Gessner et al. [168] showed the upregulation of ER stress-induced genes of the unfolded protein response (UPR) markers in the liver at one week postpartum compared to three weeks prepartum. The expression levels of these genes decreased from 1 week postpartum to later lactation. Sadri et al. [55] reported that the hepatic *TTPA* and *TAP* mRNA in dairy cows during the transition period tended to be lower than those during the peak lactation period (105 d relative to parturition), although the changes in these gene expressions were not observed during the transition period (day −21, 1 and 21 d postpartum). However, using consecutive liver tissue biopsies in the peripartum period (−4, −1, 0, 1 and 4 weeks postpartum) [2] demonstrated that *TTPA*, *AFM* and *TAP* mRNA expression were strongly downregulated immediately after calving. In the experiment, the expression of *ALB* mRNA, a negative acute-phase protein that plays the most basal hepatic function, was also downregulated, and hepatic ER stress-induced UPR and acute-phase response occurred at calving. After the first week postpartum, when the mRNA expression of *TTPA*, *AFM*, *TAP* and *ALB* recovered, the elevated UPR markers and haptoglobin mRNA expression decreased. These results suggest that α-Toc transfer from the liver into the bloodstream may be suppressed in the first days after calving because of temporal downregulation of *TTPA* and *AFM*. Ongoing research [79] showed that the hepatic mRNA expression levels of *TTPA* and *ALB* were continuously downregulated at least during the 3 d after calving. These changes in the hepatic expression of α-Toc-related genes might be associated with the maintenance of lower serum α-Toc concentrations during the first week after calving. However, there is insufficient knowledge about the hepatic expression of α-Toc-related genes and proteins in transition high-yield dairy cows. More studies are needed to delineate the relationship between hepatic α-Toc transfer and metabolism and the occurrence of hypovitaminosis E around calving.

### 4.6. Increasing Mammary α-Toc Transfer from Blood to Colostrum with Change in the Expression of α-Toc-Related Genes at Calving

It is well known that in multiparous dairy cows the α-Toc concentration in colostrum is approximately 5- to 8-fold higher than that in mature milk [13,170]. Alpha-tocopherol is a lipid-soluble micronutrient and its concentration is strongly affected by the level of milk fat; however, the α-Toc concentrations in colostrum, which were normalised by the milk fat value, were also approximately five- to eight-fold greater than those in mature milk [2,137]. The calculated α-Toc efflux with milk (concentration × milk yield) was highest in colostrum and declined in transition milk (2–3 d relative to parturition), reaching nadir levels in mature milk after one week relative to parturition [2]. In addition, the estimated mammary extraction ratios [137,171,172] of α-Toc after calving (colostrum) and at 6 weeks lactation (mature milk) were at 1.3 and 0.05%, respectively [2]. These estimations suggest that α-Toc uptake during colostrum production might be more than 20-fold greater than that in mature milk production. Furthermore, α-Toc did not accumulate in precolostrum (at one week before parturition) [2]. These results indicate that high α-Toc concentration in colostrum might be caused by the presence of a mechanism that temporarily augments a specific α-Toc transfer from the blood to colostrum across the mammary gland at calving, which might be a mechanism contributing to a lower serum α-Toc concentration at calving.

To test the possibility of this mechanism, Haga et al. [2] measured the mRNA expression levels of α-Toc-related genes in biopsied mammary gland tissues. SRBI and ABCA1 play pivotal roles in cholesterol transport, milk-fat globule synthesis and these secretions from the mammary gland [36,114,115,173]. These genes might also contribute to blood α-Toc transfer into colostrum or mature milk. However, *SCARB1* mRNA expression in mammary gland tissues was downregulated after calving. In various mouse tissues, there are SRBI-dependent and -independent pathways for tissue α-Toc uptake [95]. These observations suggest that further studies are needed to investigate the expression of other receptors, such as LDL-R and Niemann-Pick C1-Like 1 [129,173], which may be involved in α-Toc uptake in bovine mammary glands. The *ABCA1* mRNA [113,114] and protein [115] levels in bovine mammary glands during the dry-off period were higher than those during lactation. In agreement with these findings, in the transition period, *ABCA1* mRNA levels declined after calving [2]. By contrast, Mani et al. [115] demonstrated that the subcellular distribution of ABCA1 changed throughout the pregnancy-lactation cycle, and ABCA1 was present in milk-fat globule membranes isolated from fresh mature milk. It has been reported that key acceptors of cholesterol and α-Toc efflux by ABCA1, apoA1, were present in milk-fat globule membranes and the protein level was significantly higher in milk fat globule membranes prepared from colostrum than in mature milk [174]. These results suggested that ABCA1 proteins could share a function in the regulation of α-Toc transfer through localisation in the basal, apical membranes, and cytoplasm of mammary epithelial cells. The expression of *TTPA*, *SEC14L2* and *CYP4F2* mRNA in bovine mammary gland tissue and the changes in these gene expression levels during peripartum were observed [2]. The functions of αTTP and TAP in mammary epithelial cells have not been clarified, however, these results suggest that α-Toc-related genes expressed in mammary gland tissues may play an important role in the transfer of α-Toc from blood to colostrum. Further investigation to explore the physiological function of these genes in mammary glands during peripartum is needed.

A comparison of blood α-Toc profiles between mastectomized and intact dairy cows can determine the cause of decreasing blood α-Toc because the metabolic demands of colostrum production and lactation are eliminated, with only the calving effect remaining. Goff et al. [152] demonstrated that mastectomies reduced but did not eliminate loss of plasma α-Toc around calving, and strengthening the mammary transfer to colostrum does not exclusively affect the blood α-Toc concentration.

Based on recent literature, six physiological factors may be involved in α-Toc deficiency and hypovitaminosis E during the transition period of high-yielding dairy cows (Figure 2). However, the mechanisms and pathways are less well understood, and further studies are needed to understand the physiological role of α-Toc-related molecules in the GI tract and mammary gland.

## 5. Foresight

Many reviews on α-Toc and dairy cows, until Politis [30], focus primarily on the effects of supplementation on health and performance, and the utility of α-Toc as a biomarker for periparturient cow diseases has not yet been thoroughly considered. However, Qu et al. [16] indicated that lower serum α-Toc concentration is a potential early indicator for the development of LDA in multiparous cows. The same group [15] also implied that the best predictive indicators for disease were lower serum α-Toc concentrations and higher NEFA and BHBA concentrations during the prepartum period. Most recent studies on biomarkers for disease risk and milk production in periparturient dairy cows [138] performed a longitudinal, herd-based epidemiologic investigation of serum β-carotene, retinol, and α-Toc concentrations in dairy cattle on five commercial farms at three specific time points through the non-lactating and early lactation periods. The serum α-Toc concentrations, from dry-off to close-up, decreased (*p* < 0.01; LSM ± SE, 4.69 ± 1.09 to 3.00 ± 1.09 µg/mL) and then further decreased from close-up to early lactation (*p* < 0.01; 3.00 ± 1.09 to 1.44 ± 1.09 µg/mL). Higher α-Toc concentrations were associated with greater ME305 (305-d mature-equivalent milk yield), especially among cows in parity 1. Higher α-Toc concentrations were associated with decreased odds of disease among cows in parity 1, but were associated with increased odds of disease among cows in parity 2. The mechanism behind this finding is unknown, but may be associated with the increased stress cows’ parity 1 experience in the peripartum period. No vitamins were significantly associated with lameness or mastitis in multivariable models. The authors suggested that future studies should further investigate the association between serum concentrations of lipid vitamins and periparturient cow diseases to establish serum ranges at which these biomarkers indicate increased disease risk. According to a recent study [175], whole blood samples can be directly used, and the measurement of lipid vitamin levels can be performed effortlessly in less than 5 min, even at the cow-side, using a field-portable fluorometer/spectrophotometer (iCheck) without further sample preparation. Based on this new development, the concentrations of VE, β-carotene, and vitamin A in the blood can be used as nutritional biomarkers to directly optimise nutritional interventions at the farm, together with stakeholders such as veterinarians, farmers, nutritional advisors, and feed consultants. Thus, the wave of the future, using serum vitamin concentrations as biomarkers for disease risk during the periparturient period, in addition to monitoring dietary supplementation, may prove to be an effective tool for improving animal health. Furthermore, a better understanding of the physiological factors underlying the cause of hypovitaminosis E in the transition period will improve the utility of α-Toc as a biomarker for periparturient high-yield dairy cow diseases.

## 6. Conclusions

Numerous studies and reviews about the disease risk of hypovitaminosis E (<3 μg/mL) and the effect of α-Toc supplementation on the health and performance of transition dairy cows and heifers have been published in the last 30 years. However, the risk and supplemental effects are controversial because several factors affect the availability of α-Toc and its physiological functions in cows. In the current review, we focused on relevant papers published in the last 10 years and presented some novel data about the disease risk of hypovitaminosis E and the effects of α-Toc supplementation in transition dairy cows. These data strongly demonstrate that hypovitaminosis E in the transition period is a risk factor for the occurrence of peripartum disease and lower performance in dairy cows. Alpha-tocopherol supplementation of more than 3000 IU/day during the prepartum period can be important for the effective prevention of peripartum disease in high-yield dairy cows. Furthermore, a study on the effectiveness of using serum vitamin levels as biomarkers to predict disease in dairy cows was reported, and a rapid field test (cow-side assay) for measuring vitamin levels using whole blood was developed. By contrast, evidence for how hypovitaminosis E occurred during the transition period was scarce until the 2010s. Pioneering studies conducted with humans and rodents have identified and characterised some α-Toc-related proteins, molecular players involved in α-Toc regulation, from the 1990s, followed by a study in ruminants from the 2010s. Based on the recent literature, six physiological factors may be involved in α-Toc deficiency and hypovitaminosis E during the transition period of high-yielding dairy cows. However, the mechanisms and pathways are less well understood, and further studies are needed to understand the physiological role of α-Toc-related molecules in cattle. In the future, understanding the molecular mechanisms underlying hypovitaminosis E will contribute to the prevention of peripartum disease and high performance in dairy cows.

## Figures and Tables

**Figure 1 animals-11-01088-f001:**
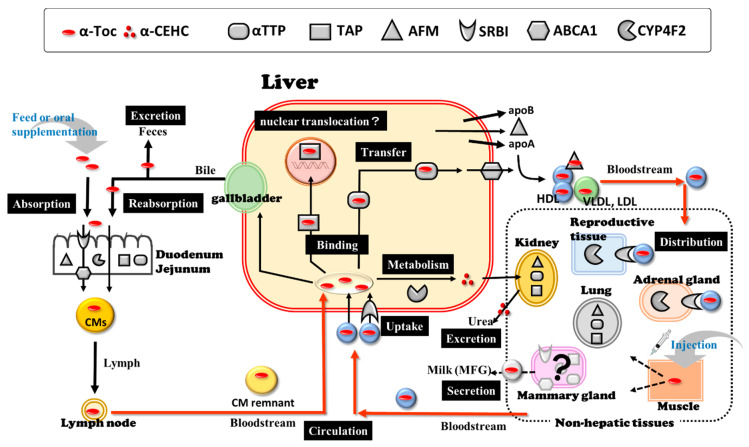
The possible mechanism of α-Toc disposition (metabolism, transportation, and tissue distribution) with the expression of α-Toc-related genes in cattle. Abbreviations: α-Toc, α-tocopherol; α-CEHC, α-carboxyethyl hydroxychromans; αTTP, α-tocopherol transfer protein; TAP, tocopherol associated protein; AFM, afamin; SRBI, scavenger receptor class B, Type I; ABCA1, ATP-binding cassette transporter A1; CYP4F2, cytochrome P450 family 4, subfamily F, polypeptide 2; CM, chylomicron; HDL, high density lipoprotein; VLDL, very low-density lipoprotein; LDL, low density lipoprotein; MFG, milk fat globules.

**Figure 2 animals-11-01088-f002:**
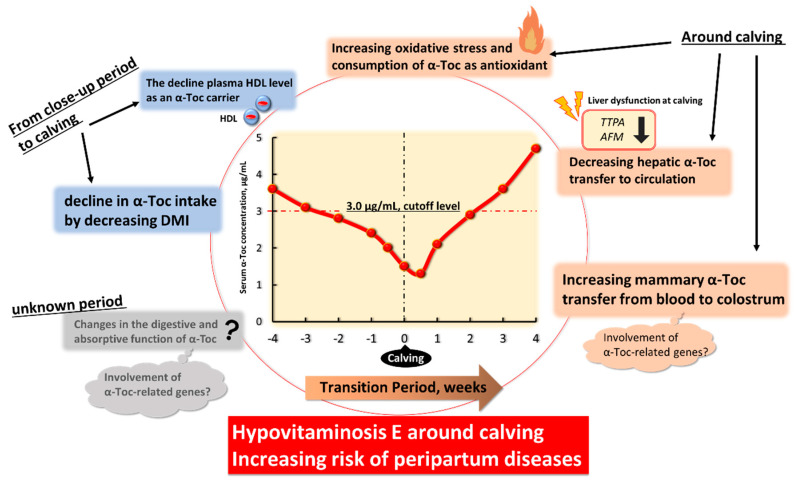
Six candidate physiological factors causing decreased blood α-tocopherol (α-Toc) level and hypovitaminosis E in transition high-yield dairy cows. Abbreviations: DMI, dry matter intake; *TTPA*, α-tocopherol transfer protein gene; *AFM*, afamin gene; HDL, high density lipoprotein.

## Data Availability

Not applicable.

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
