# Peer review of "The Physiological Roles of Vitamin E and Hypovitaminosis E in the Transition Period of High-Yielding Dairy Cows"

_animals, 2021, doi:10.3390/ani11041088_

Round 1
Reviewer 1 Report
During the transition period, high-yield dairy cows experience severe energy and nutrient deficiencies. It is the key to the health of the cows.
Because, the ruminal fermentation indices should also provided. This may influence for the quality of animal products origin and animal health.
In addition, the "Simple Summary" section is too long and does not conform to the guidelines.
Author Response
Please read PDF file.

Reviewer 2 Report
This is a very comprehensive review of the role of Vitamin E in dairy cows shortly before and after calving. Current knowledge is covered well with some suggestions for further study.
Minor issues
Line 52. “Transition period” is not defined. I have seen it referred to as the 3 weeks before and after calving, but also as the period 2 months before to 1 month after calving. A definition would be useful.
Line 128. Delete “only”
Line 338-340. “As if it were common sense in experiment on farms from 2000s to 2020s, the occurrence of hypovitaminosis E around calving has been still reported [2,13,15,16,79,135-138].”
This sentence needs to be reworded, as it is not clear what it means.
Line 489. Change “increase” to “increases”
Line 501. Change “0.5 week relative to parturition” to “0.5 weeks after parturition”.
Figure 2. “circuration” should probably be “circulation”
Author Response
Please read PDF file.
